

# The boundary disorder correlation for the Ising model on a cylinder

**Rafael Leon Greenblatt**

Dipartimento di Matematica, Università degli Studi di Roma "Tor Vergata", Rome, Italy

greenblatt@mat.uniroma2.it

## Abstract

**I give an expression for the correlation function of disorder insertions on the edges of the critical Ising model on a cylinder as a function of the aspect ratio (rescaled in the case of anisotropic couplings). This is obtained from an expression for the finite size scaling term in the free energy on a cylinder in periodic and antiperiodic boundary conditions in terms of Jacobi theta functions.**

## 1 Introduction

I study the disorder correlation

$$\left\langle \mu_{\text{top}}\mu_{\text{bottom}} \right\rangle_{\pm} := \frac{Z_{\mp}}{Z_{\pm}}, \tag{1}$$

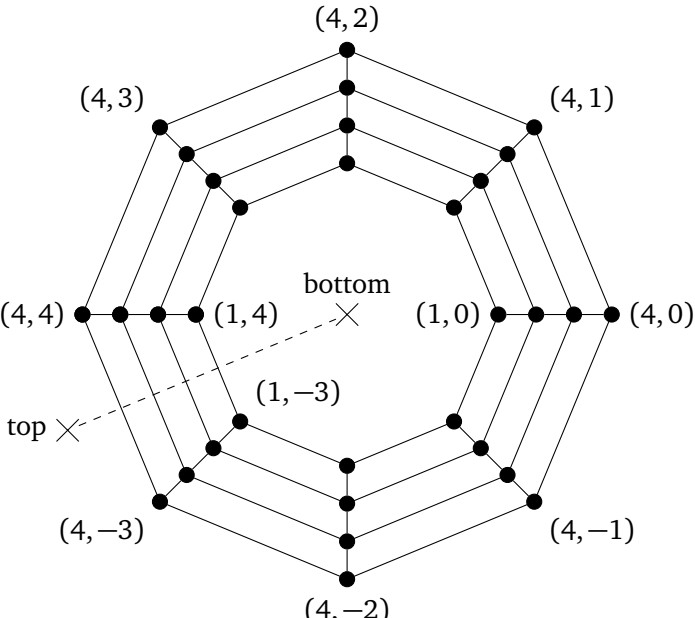

Figure 1: An example of the lattice under consideration with $\mathcal{M} = 2, \mathcal{N} = 4$, drawn as a planar graph. Values of the indices $(j, k)$ are shown for some of the sites. The faces corresponding to the disorder insertions in Equation (1) are labelled, along with the path between them which crosses the bonds corresponding to the $\pm$ sign in Equation (2).

where $Z_+$ ($Z_-$) is the partition function of the Ising model on a discrete $2\mathcal{M} \times 2\mathcal{N}$ cylinder with open and (anti-)periodic boundary conditions, that is

$$Z_\pm = \sum_{\sigma \in \{-1,+1\}^{4\mathcal{M}\mathcal{N}}} \exp\left(\sum_{j=1}^{2\mathcal{M}} \sum_{k=-\mathcal{N}+1}^{\mathcal{N}-1} \beta E_1 \sigma_{j,k}\sigma_{j,k+1} + \sum_{j=1}^{2\mathcal{M}-1} \sum_{k=-\mathcal{N}+1}^{\mathcal{N}} \beta E_2 \sigma_{j,k}\sigma_{j+1,k}\right. \quad (2)$$
$$\left. \pm \sum_{j=1}^{2\mathcal{M}} \beta E_1 \sigma_{j,\mathcal{N}}\sigma_{j,-\mathcal{N}+1}\right),$$

with $j = 1, \ldots, 2\mathcal{M}$ the coordinate in the nonperiodic direction, and $k = -\mathcal{N} + 1, \ldots, \mathcal{N}$ the coordinate in the (anti-)periodic direction; this partition function does not depend on the signs of $E_1, E_2$ so I will assume them to be positive. This lattice can be embedded in the plane as a graph with two faces corresponding to the boundaries of the cylinder (see Figure 1), and reversing the sign of the bonds crossed by a path between the two faces exchanges $Z_+$ and $Z_-$, so this corresponds to the notion of disorder insertions introduced by Kadanoff and Ceva [1].

This quantity appears when expressing the correlation function in periodic boundary conditions for an odd number of spins on each boundary. In the simplest case, the Kadanoff-Ceva Fermion two-point function in antiperiodic boundary conditions given by

$$\left\langle \psi_{x,\text{top}} \psi_{y,\text{bottom}} \right\rangle_- := \left\langle \sigma_x \mu_{\text{top}} \sigma_y \mu_{\text{bottom}} \right\rangle_- = \left\langle \sigma_x \sigma_y \right\rangle_+ \left\langle \mu_{\text{top}} \mu_{\text{bottom}} \right\rangle_- . \quad (3)$$

Kadanoff-Ceva correlation functions at different positions satisfy a system of linear equations [2] which allow them to be characterized as the solution of a linear boundary value problem [3] and evaluated via a modified Fourier transform, giving a particularly explicit form in the continuum limit [4].

Higher spin correlations can be calculated from

$$\left\langle \psi_{x_1,\text{top}} \ldots \psi_{x_m,\text{top}} \psi_{y_1,\text{bottom}} \ldots \psi_{y_n,\text{bottom}} \right\rangle_- = \left\langle \sigma_{x_1} \ldots \sigma_{x_m} \sigma_{y_1} \ldots \sigma_{y_n} \right\rangle_+ \left\langle \mu_{\text{top}} \mu_{\text{bottom}} \right\rangle_- \quad (4)$$

(here for $m, n$ odd, using $\mu_{\text{top}}^2 = \mu_{\text{bottom}}^2 = 1$ to simplify), together with the fact that the correlation function on the left-hand side can be expressed using similar linear relationships [3] as the Pfaffian of a matrix whose elements are values of the two-point function in Equation (3) or

$$\left\langle \psi_{x,\text{top}} \psi_{x',\text{top}} \right\rangle_- = \left\langle \sigma_x \sigma_{x'} \right\rangle_- , \quad \left\langle \psi_{y,\text{bottom}} \psi_{y',\text{bottom}} \right\rangle_- = \left\langle \sigma_y \sigma_{y'} \right\rangle_- . \tag{5}$$

The result is an expression for the multi-point boundary spin correlation function (with periodic boundary conditions) as a Pfaffian in terms of the two-point boundary spin correlation with either periodic or antiperiodic boundary conditions and the disorder correlation of Equation (1), see [5]. Note that this is slightly different from the situation for with an even number of spins on each boundary, where (as for simply connected domains [6]) the boundary spin correlation function is immediately equal to a Kadanoff-Ceva correlation function and so is given by a Pfaffian in terms of the two point spin function in the same boundary conditions, without any additional factor.

At the critical temperature in the limit of $\mathcal{M}, \mathcal{N} \to \infty$ with $\mathcal{M}/\mathcal{N}^2 \to 0$, I will show that

$$\left\langle \mu_{\text{top}} \mu_{\text{bottom}} \right\rangle_\pm \sim \left( \frac{\theta_2(0, e^{-2\pi\zeta})}{2\theta_3(0, e^{-2\pi\zeta})} \right)^{\pm 1/2} = \left( \frac{\theta_2(0|2i\zeta)}{2\theta_3(0|2i\zeta)} \right)^{\pm 1/2} , \tag{6}$$

where $\theta_j$ are Jacobi theta functions [7, Chapter 20] (I follow the notational conventions used there, in particular to distinguish the two parameterizations of the functions which are related by $\theta_j(z|\tau) = \theta_j(z, q)$ whenever $q = e^{i\pi\tau}$), and $\zeta = \Xi \mathcal{M}/\mathcal{N}$ is a rescaled aspect ratio with a factor $\Xi$, introduced in Equation (14) and depending only on the ratio of coupling constants $E_1/E_2$ (cf. [8]), with $\Xi = 1$ in the isotropic case $E_1 = E_2$. Note that all of the dependence of this asymptotic form on the shape of the lattice and on the coupling strengths is included in this parameter.

Equation (6) follows from an asymptotic expansion of the logarithm of the partition functions at the critical temperature to constant order, i.e.

$$\log Z_\pm = p\mathcal{M}\mathcal{N} + s\mathcal{N} + z_\pm(\zeta) + \mathcal{O}(1/\mathcal{N}) + \mathcal{O}(\mathcal{M}/\mathcal{N}^2), \tag{7}$$

where $p$ and $s$ are unchanged between the periodic and antiperiodic cases; thus the constant-order term $z_\pm$ (also called the finite size scaling term) is the first term which differs between the two cases, and so it entirely describes the leading behaviour of the disorder correlation in Equation (1). The presence of a term of this order for the Ising model was first noted by Ferdinand and Fisher [9] for the torus, who also gave an expression in terms of theta functions. It was subsequently noted that the behaviour of such terms (including at least some of their dependence on boundary conditions) could be explained by the identification of the scaling limit of the Ising model with a specific conformal field theory [10–12], stimulating a number of further calculations in other geometries and boundary conditions, e.g. [13–16]. For the periodic cylindrical case this expansion was first discussed in [17] although only part of the result was presented without the details of the calculation. A fuller calculation (for the periodic case $Z_+$ with $\Xi = 1$, i.e. isotropic couplings) was presented by the present author in the unpublished preprint [18]; in what follows I adapt and expand the relevant and novel parts of that calculation. A similar expansion has been carried out in [19], giving a less explicit expression which however is also valid away from the critical temperature (as in a massive scaling limit), in both periodic and antiperiodic boundary conditions. Expansions similar to Equation (7) have also been obtained for the dimer model in a variety of geometries (e.g. [20–22]), including some cases giving formulae for similar ratios of partition functions [23–25].

## 2 The partition function in the McCoy-Wu solution

The starting point is the exact solution of the Ising model on a cylinder due to McCoy and Wu [26, Chapter VI, particularly p. 113-20], which starts from the expression of the high-temperature contour representation as a dimer model on an augmented lattice and then takes advantage of the (anti-)periodicity in one direction to reduce the two-dimensional problem to a collection of one-dimensional problems formulated in terms of recursion relationships expresses the partition function in terms of products of $2 \times 2$ matrices similar to the transfer matrix of the one-dimensional Ising model, also used in the study of disordered models where these become random matrices [27–29], and can be generalized (not without complications) to a rectangle with open boundary conditions [30, 31]. For brevity, I use the resulting formulae from the McCoy-Wu solution as a starting point without discussing their derivation in detail.

Since the solution is based on the high-temperature contour expansion, the main parameters are

$$z_i := \tanh \beta E_i, \quad i = 1, 2, \tag{8}$$

and the critical temperature is characterized by

$$z_2 = \frac{1 - z_1}{1 + z_1} \tag{9}$$

(this is a rearranged version of [26, Chapter V, Equation (3.9)]; it reduces to $z_1 = z_2 = \sqrt{2} - 1$ in the isotropic case $E_1 = E_2 \implies z_1 = z_2$), which I will sometimes use below to eliminate $z_2$. Using commonplace identities about hyperbolic functions, Equation (9) can be rewritten as

$$\sinh 2\beta E_1 \ \sinh 2\beta E_2 = 1, \tag{10}$$

from which it is clear that the critical value of $\beta$ (and so of $z_1, z_2$) depends only on the ratio $E_1/E_2$.

The calculation [26, Chapter VI], which were given for periodic boundary conditions, generalize easily to antiperiodic boundary conditions, along the lines of the calculation on the torus in [26, Section IV.6]. Note that McCoy and Wu also allowed an additional term in the Hamiltonian coupling to the sites on one of the boundaries, but we take this term to be zero (setting $\mathfrak{H} = 0$).

The upshot of this is

$$Z_\pm^2 = (2 \cosh \beta E_1)^{8\mathcal{M}\mathcal{N}} (\cosh \beta E_2)^{4\mathcal{N}(2\mathcal{M}-1)} \prod_{\theta \in Q_\pm(\mathcal{N})} \left\{ \left| 1 + z_1 e^{i\theta} \right|^{4\mathcal{M}} \lambda^{2\mathcal{M}} \left[ v^2 + v'^2 \alpha^{-4\mathcal{M}} \right] \right\} \tag{11}$$

(cf. [26, Chapter 6, Equation (3.26)]; as noted above $\mathfrak{H} = 0$, so $z = 0$, giving a slight simplification), where the product is over

$$Q_+(\mathcal{N}) = \left\{ \frac{\pi(2n-1)}{2\mathcal{N}} : n = 1, \dots, 2\mathcal{N} \right\}, \quad \text{or} \quad Q_-(\mathcal{N}) = \left\{ \frac{\pi n}{\mathcal{N}} : n = 0, \dots, 2\mathcal{N} - 1 \right\}, \tag{12}$$

according to the boundary condition, and

$$\lambda = \frac{z_2(1 - z_1^2)}{|1 + z_1 e^{i\theta}|^2} \alpha, \tag{13}$$

and specializing [26, Chapter VI, Equation (3.16)] to the critical case using Equation (9)

to eliminate $z_2$,

$$
\begin{aligned}
\alpha &= \frac{1}{2(1-z_1)^2} \left\{ 2\frac{(1+z_1^2)^2}{(1+z_1)^2} - \frac{4z_1^2}{(1+z_1)^2}(e^{i\theta}+e^{-i\theta}) \right. \\
&\qquad\qquad \left. + \frac{4z_1}{(1+z_1)^2}\left[(1-z_1^2 e^{i\theta})(1-z_1^2 e^{-i\theta})(1-e^{i\theta})(1-e^{-i\theta})\right]^{1/2} \right\} \\
&= \frac{1}{(1-z_1^2)^2}\left\{(1+z_1^2)^2 - 4z_1^2\cos\theta + 2z_1\left[2(1-\cos\theta)(1+z_1^4 - 2z_1^2\cos\theta)\right]^{1/2}\right\} \\
&= 1 + \frac{2z_1}{1-z_1^2}|\theta| + \left(\frac{z_1}{1-z_1^2}\right)^2\theta^2 + \mathcal{O}(\theta^3) = \exp\left(\frac{2z_1}{1-z_1^2}|\theta|\right) + \mathcal{O}(\theta^3) \\
&=: \exp(\Xi|\theta|) + \mathcal{O}(\theta^3), \quad \theta\to 0
\end{aligned}
\tag{14}
$$

(in particular [26, Chapter 6, Equation (3.17)] - the expression for two of the coefficients appearing in this equation - simplifies under the criticality condition to $\alpha_1 = z_1^2, \alpha_2 = 1$); and $v, v' \geq 0$ are such that

$$
v^2 + v'^2 = 1, \quad \frac{v}{v'} = \frac{i(z_2^2 - \lambda)}{\mathfrak{a}z_2}, \tag{15}
$$

with

$$
\mathfrak{a} = \frac{2iz_1\sin\theta}{|1+z_1 e^{i\theta}|^2}, \tag{16}
$$

so that

$$
\frac{v'}{v} = \frac{2z_1\sin\theta}{z_2|1+z_1 e^{i\theta}|^2 - (1-z_1^2)\alpha}, \qquad v^2 = \left(1+\frac{v'^2}{v^2}\right)^{-1}. \tag{17}
$$

## 3 Asymptotic analysis

In order to study the behavior of the partition function in the limit of interest $\mathcal{M},\mathcal{N}\to\infty$, $\mathcal{M}/\mathcal{N}^2 \to 0$, I separate the right hand side of Equation (11) into three factors as

$$
Z_\pm = Y(\mathcal{M},\mathcal{N})\prod_{\theta\in Q_\pm(\mathcal{N})} W_\mathcal{M}(\theta)X_\mathcal{M}(\theta), \tag{18}
$$

with

$$
Y(\mathcal{M},\mathcal{N}) = (2\cosh\beta E_1)^{4\mathcal{M}\mathcal{N}}(\cosh\beta E_2)^{2\mathcal{N}(2\mathcal{M}-1)}z_2^{\mathcal{M}\mathcal{N}}(1-z_1^2)^{\mathcal{M}\mathcal{N}}, \tag{19}
$$

$$
W_\mathcal{M}(\theta) = \alpha^\mathcal{M}|v|, \tag{20}
$$

$$
X_\mathcal{M}(\theta) = \left[1 + \frac{v^2}{v'^2}\alpha^{-4\mathcal{M}}\right]^{1/2}. \tag{21}
$$

The first factor is clearly independent of the boundary condition $\pm$, and its dependence on the system size is perfectly straightforward, so that

$$
\log Y(\mathcal{M},\mathcal{N}) = p_1\mathcal{M}\mathcal{N} + s_1\mathcal{N}, \tag{22}
$$

with $p_1, s_1$ independent of the boundary condition.

For the other two parts, the behavior of various quantities near $\theta = 0$ is particularly important. It is easy enough to see from Equation (14) that $\alpha$ depends smoothly on $\theta \in (0, 2\pi)$ but its derivative is discontinuous at zero, with

$$
\frac{d}{d\theta}\log\alpha(\theta) = \pm\Xi + \mathcal{O}(\theta^2), \quad \theta\to 0^\pm. \tag{23}
$$

With this in mind, we use the Euler-MacLaurin formula [7, Section 2.10(i)] to note that

$$
\begin{aligned}
\sum_{\theta \in Q_+(\mathcal{N})} f(\theta) &= \sum_{n=1}^{2\mathcal{N}} f\left(\frac{\pi(2n-1)}{2\mathcal{N}}\right) \\
&= \frac{\mathcal{N}}{\pi} \int_{\pi/2\mathcal{N}}^{(2-1/2\mathcal{N})\pi} f(\theta)\, d\theta + \tfrac{1}{2} f(\pi/2\mathcal{N}) + \tfrac{1}{2} f((2-1/2\mathcal{N})\pi) \\
&\quad + \frac{1}{12}\frac{\pi}{\mathcal{N}}\left[f'((2-1/2\mathcal{N})\pi) - f'(\pi/2\mathcal{N})\right] + \mathcal{O}\left(\frac{1}{\mathcal{N}^2}\right) \\
&= \frac{\mathcal{N}}{\pi} \int_0^{2\pi} f(\theta)\, d\theta + \frac{1}{24}\frac{\pi}{\mathcal{N}}\left[f'(0+) - f'(2\pi-)\right] + \mathcal{O}\left(\frac{1}{\mathcal{N}^2}\right), \quad \mathcal{N} \to \infty,
\end{aligned}
\tag{24}
$$

for any $f$ which is three times differentiable on $(0, 2\pi)$ with bounded third derivative, using

$$
\begin{aligned}
\frac{\mathcal{N}}{\pi} \int_0^{\pi/2\mathcal{N}} f(\theta)\, d\theta &= \frac{1}{2} f(\pi/2\mathcal{N}) - \int_0^{\pi/2\mathcal{N}} \int_x^{\pi/2\mathcal{N}} f'(y)\, dy\, dx \\
&= \frac{1}{2} f(\pi/2\mathcal{N}) - \frac{1}{8}\frac{\pi}{\mathcal{N}} f'(0+) + \mathcal{O}\left(\frac{1}{\mathcal{N}^2}\right), \quad \mathcal{N} \to \infty,
\end{aligned}
\tag{25}
$$

to obtain the last expression; combining Equation (24) with Equation (23) gives

$$
\sum_{\theta \in Q_+(\mathcal{N})} \log \alpha(\theta) = \frac{\mathcal{N}}{\pi} \int_0^{2\pi} f(\theta)\, d\theta + \frac{\zeta}{12}\frac{\pi}{\mathcal{N}} + \mathcal{O}\left(\frac{1}{\mathcal{N}^2}\right), \quad \mathcal{N} \to \infty.
\tag{26}
$$

Similarly, assuming $f(0) = f(2\pi)$

$$
\begin{aligned}
\sum_{\theta \in Q_-(\mathcal{N})} f(\theta) &= \sum_{n=0}^{2\mathcal{N}} f\left(\frac{n\pi}{\mathcal{N}}\right) - \tfrac{1}{2} f(0) - \tfrac{1}{2} f(2\pi) \\
&= \frac{\mathcal{N}}{\pi} \int_0^{2\pi} f(\theta)\, d\theta - \frac{1}{12}\frac{\pi}{\mathcal{N}}\left[f'(0^+) - f'(2\pi-)\right] + \mathcal{O}\left(\frac{1}{\mathcal{N}^2}\right), \quad \mathcal{N} \to \infty,
\end{aligned}
\tag{27}
$$

which combines with Equation (23) to give

$$
\sum_{\theta \in Q_-(\mathcal{N})} \log \alpha(\theta) = \frac{\mathcal{N}}{\pi} \int_0^{2\pi} f(\theta)\, d\theta - \frac{\zeta}{6}\frac{\pi}{\mathcal{N}} + \mathcal{O}\left(\frac{1}{\mathcal{N}^2}\right), \quad \mathcal{N} \to \infty.
\tag{28}
$$

Equations (26) and (28) can be summarized together, noting that both contain the same integral, as

$$
\sum_{\theta \in Q_\pm(\mathcal{N})} \log \alpha(\theta) = p_2 \mathcal{N} + \frac{w_\pm}{\mathcal{N}} + \mathcal{O}\left(\frac{1}{\mathcal{N}^2}\right), \quad \mathcal{N} \to \infty,
\tag{29}
$$

with $p_2$ independent of the boundary condition and

$$
w_+ = \frac{\pi}{12}\Xi, \quad w_- = -\frac{\pi}{6}\Xi,
\tag{30}
$$

corresponding to the coefficients of $1/\mathcal{N}$ evaluated with $f = \log \alpha$ using Equation (23).

On the other hand it is clear from Equation (17) that $\log|v| = \log|v(\theta)|$ is continuously differentiable as a function of $\theta$, so that Equations (24) and (27) give

$$
\sum_{\theta \in Q_\pm(\mathcal{N})} \log|v| = \frac{\mathcal{N}}{\pi} \int_0^{2\pi} \log|v(\theta)|\, d\theta + \mathcal{O}\left(\frac{1}{\mathcal{N}^2}\right), \quad \mathcal{N} \to \infty,
\tag{31}
$$

with only the remainder depending on the boundary condition.

Combining Equations (22), (29) and (31)

$$\log Y(\mathcal{M},\mathcal{N}) + \sum_{\theta \in Q_\pm(\mathcal{N})} \log W_\mathcal{M}(\theta) = p\mathcal{M}\mathcal{N} + s\mathcal{N} + w_\pm \frac{\mathcal{M}}{\mathcal{N}} + \mathcal{O}\left(\frac{1}{\mathcal{N}} + \frac{\mathcal{M}}{\mathcal{N}^2}\right), \qquad (32)$$

for $\mathcal{M},\mathcal{N} \to \infty, \mathcal{M}/\mathcal{N}^2 \to 0$, where $p$ and $s$ are coefficients independent of the boundary condition.

For the remaining factor, first note that the contribution from most of $Q_\pm(\mathcal{N})$ can be bounded as

$$\sum_{\theta \in Q_\pm(\mathcal{N}) \cap [\mathcal{M}^{-1/3}, 2\pi - \mathcal{M}^{-1/3}]} \log X_\mathcal{M}(\theta) = \mathcal{O}\left(\sum_{\theta \in Q_\pm(\mathcal{N}) \cap [\mathcal{M}^{-1/3}, 2\pi - \mathcal{M}^{-1/3}]} \alpha(\theta)^{-4\mathcal{M}}\right)$$

$$\qquad (33)$$

$$= \mathcal{O}\left(\sum_{n=\lfloor \mathcal{N}\mathcal{M}^{-1/3}\rfloor}^{\infty} e^{-c(\mathcal{M}/\mathcal{N})n}\right) = \mathcal{O}\left(\frac{\mathcal{N}}{\mathcal{M}} e^{-c\mathcal{M}^{2/3}}\right),$$

for some $c > 0$, which is negligible; this sum excludes only terms with small $\theta$, which can be approximated by a simpler expression, beginning with the bound

$$\left|\log X_\mathcal{M}^2(\theta) - \log(1 + e^{-4\Xi\mathcal{M}\theta})\right| \le \left|\left(\frac{v'}{v}\right)^2 \alpha^{-4\mathcal{M}} - e^{-4\Xi\mathcal{M}|\theta|}\right|$$

$$\qquad (34)$$

$$\le \left|\left(\frac{v'}{v}\right)^2 - 1\right| \alpha^{-4\mathcal{M}} + \left|\alpha^{-4\mathcal{M}} - e^{-4\Xi\mathcal{M}|\theta|}\right|.$$

Using the asymptotic expansion in Equation (14) in Equation (17) gives

$$\frac{v'}{v} = \frac{2z_1\theta + \mathcal{O}(\theta^2)}{\alpha - 1 + \mathcal{O}(\theta^2)} = -\frac{\theta}{|\theta|} + \mathcal{O}(\theta) \implies \left(\frac{v'}{v}\right)^2 = 1 + \mathcal{O}(\theta), \quad \theta \to 0, \qquad (35)$$

and again using the asymptotics from Equation (14) this gives

$$\left|\left(\frac{v'}{v}\right)^2 - 1\right| \alpha^{-4\mathcal{M}} \le C\theta e^{-\mathcal{M}\theta}, \quad |\theta| \le \mathcal{M}^{-1/3}; \qquad (36)$$

also

$$\left|\alpha^{-4\mathcal{M}} - e^{-4\Xi\mathcal{M}\theta}\right| \le 4\mathcal{M}\left|\alpha - e^{\Xi\theta}\right|\left[\min(\alpha, e^{\Xi\theta})\right]^{-4\mathcal{M}-1} \le C\mathcal{M}\theta^2 e^{-c\mathcal{M}\theta}, \qquad (37)$$

and putting all this together (note that $X_\mathcal{M}$ is even in $\theta$ with $X_\mathcal{M}(0) = 1$)

$$\sum_{\theta \in Q_+(\mathcal{N})} \log X_\mathcal{M}(\theta) - \log \prod_{n=1}^{\infty}\left[1 + \exp\left(-2\pi\Xi\frac{\mathcal{M}}{\mathcal{N}}(2n-1)\right)\right]$$

$$\qquad (38)$$

$$= \mathcal{O}\left(\mathcal{N}\int_0^\infty (k + \mathcal{M}k^2)e^{-k\mathcal{M}}\,\mathrm{d}k\right) = \mathcal{O}\left(\frac{\mathcal{N}}{\mathcal{M}^2}\right),$$

$$\sum_{\theta \in Q_-(\mathcal{N})} \log X_\mathcal{M}(\theta) - \log \prod_{n=1}^{\infty}\left[1 + \exp\left(-4\pi\Xi\frac{\mathcal{M}}{\mathcal{N}}n\right)\right]$$

$$\qquad (39)$$

$$= \mathcal{O}\left(\mathcal{N}\int_0^\infty (k + \mathcal{M}k^2)e^{-k\mathcal{M}}\,\mathrm{d}k\right) = \mathcal{O}\left(\frac{\mathcal{N}}{\mathcal{M}^2}\right).$$

As noted in [9], the products appearing here can be expressed in terms of Jacobi theta functions [7, Chapter 20, especially Equations (20.5.3–4)] as

$$\prod_{n=1}^{\infty} \left[1 + \exp\left(-2\pi\zeta(2n-1)\right)\right]^2 = \frac{\theta_3(e^{-2\pi\zeta})}{\theta_0(e^{-2\pi\zeta})}, \tag{40}$$

$$\prod_{n=1}^{\infty} \left[1 + \exp\left(-4\pi\zeta n\right)\right]^2 = e^{\frac{\pi}{2}\Xi\frac{M}{N}} \frac{\theta_2(e^{-2\pi\zeta})}{2\theta_0(e^{-2\pi\zeta})}, \tag{41}$$

where $\zeta = \Xi\mathcal{M}/\mathcal{N}$ and $\theta_j(q)$ is an abbreviation for $\theta_j(0,q)$, and

$$\theta_0(q) := \prod_{n=1}^{\infty}(1-q^{2n}) = q^{-1/12}\left[\tfrac{1}{2}\theta_2(q)\theta_3(q)\theta_4(q)\right]^{1/3}, \tag{42}$$

see [7, Equations (20.4.6), (23.15.9), (23.17.8)].

Collecting all this,

$$\sum_{\theta \in Q_{\pm}(\mathcal{N})} \log X_{\mathcal{M}}(\theta) = z_{\pm}(\zeta) - w_{\pm}\frac{\mathcal{M}}{\mathcal{N}} + \mathcal{O}\left(\frac{\mathcal{N}}{\mathcal{M}^2}\right), \tag{43}$$

where $w_{\pm}$ is as in Equation (30) (leading to a cancellation with the corresponding term in Equation (29)) and

$$z_+(\zeta) = \frac{1}{6}\log\left(\frac{2\theta_3^2(e^{-2\pi\zeta})}{\theta_2(e^{-2\pi\zeta})\theta_4(e^{-2\pi\zeta})}\right), \quad z_-(\zeta) := \frac{1}{6}\log\left(\frac{\theta_2^2(e^{-2\pi\zeta})}{4\theta_3(e^{-2\pi\zeta})\theta_4(e^{-2\pi\zeta})}\right). \tag{44}$$

Equation (7) then follows by substituting Equations (32) and (43) into Equation (18). Substituting this in turn into Equation (1) gives

$$\left\langle\mu_{\text{top}}\mu_{\text{bottom}}\right\rangle_{\pm} = \exp\left(z_{\mp}(\zeta) - z_{\pm}(\zeta)\right) + \mathcal{O}(\mathcal{M}/\mathcal{N}^2) + \mathcal{O}(1/\mathcal{M}), \tag{45}$$

for $\mathcal{M}, \mathcal{N} \to \infty$, $\mathcal{M}/\mathcal{N}^2 \to 0$. Then Equation (6) follows by inserting Equation (44).

## 4 Comparison to the predictions of conformal field theory

As a check on the calculation we can compare the asymptotics of this finite size scaling term with the predictions of [11], as has already been done for the corresponding terms in a number of other boundary conditions [13,17] and for related expressions for cylindrical models [8,19]. At least in the isotropic case ($\Xi = 1$, hence $\zeta = \mathcal{M}/\mathcal{N}$), the predictions for periodic and antiperiodic boundary conditions (sectors P and A in [11, Table 1]; note that there is a sign difference, since the formulae in [11] refer to the free energy $-\log Z$) correspond in the setup of this paper to

$$z_+(\zeta) \sim \frac{\pi}{12}\zeta, \qquad z_-(\zeta) \sim -\frac{\pi}{6}\zeta, \tag{46}$$

asymptotically in the limit of an infinitely long tube ($\mathcal{M}, \mathcal{N} \to \infty$ with $\mathcal{M} >> \mathcal{N}$, hence $\zeta \to \infty$), while the prediction for open (free) boundary conditions gives

$$z_{\pm}(\zeta) \sim \frac{\pi}{48}\zeta^{-1}, \tag{47}$$

for the limit of an infinite strip ($\mathcal{N} >> \mathcal{M}$, $\zeta \to 0^+$), where the boundary condition indicated by $\pm$ disappears.

To compare this with the results of the previous section (when it applies, so stipulating that the limits are taken with $\mathcal{M}/\mathcal{N}^2 \to 0$) first requires an asymptotic expansion of Equation (44), which can be carried out with the series representations [7, Equations (20.2.2)–(20.2.4)] of the theta functions, which give

$$\theta_2(q) \sim 2q^{1/4}\,, \tag{48}$$

$$\theta_3(q) \sim 1\,, \tag{49}$$

$$\theta_4(q) \sim 1\,, \tag{50}$$

all for $q \to 0$, which corresponds to the limit $\zeta \to \infty$ (i.e. $\mathcal{M} >> \mathcal{N}$) of Equation (44), where the theta functions appear with $q := e^{-2\pi\zeta}$. Consequently

$$z_+(\zeta) \sim \frac{1}{6}\log q^{-1/4} = \frac{\pi}{12}\zeta\,, \quad z_-(\zeta) \sim \frac{1}{6}\log q^{1/2} = -\frac{\pi}{6}\zeta\,, \tag{51}$$

matching Equation (46). For the limit $\zeta \to 0$, we use Jacobi's imaginary transformation [7, Equations (20.7.31-33)]; note the use of the parameter $\tau$ with $q = e^{i\pi\tau}$, so $\tau = 2i\zeta$ and $\tau' = 1/\tau = i/2\zeta$), which for the quantities at hand takes the form

$$\theta_2(e^{-\pi x}) = x^{-1/2}\theta_4(e^{-\pi/x})\,, \tag{52}$$

$$\theta_3(e^{-\pi x}) = x^{-1/2}\theta_3(e^{-\pi/x})\,, \tag{53}$$

$$\theta_4(e^{-\pi x}) = x^{-1/2}\theta_2(e^{-\pi/x})\,, \tag{54}$$

to rewrite Equation (44) so that the theta functions once again appear with small arguments as

$$z_+(\zeta) = \frac{1}{6}\log\left(\frac{2\theta_3^2\left(e^{-\pi/2\zeta}\right)}{\theta_2\left(e^{-\pi/2\zeta}\right)\theta_4\left(e^{-\pi/2\zeta}\right)}\right) \sim \frac{\pi}{48}\zeta^{-1}\,, \tag{55}$$

$$z_-(\zeta) = \frac{1}{6}\log\left(\frac{\theta_4^2\left(e^{-\pi/2\zeta}\right)}{4\theta_3\left(e^{-\pi/2\zeta}\right)\theta_4\left(e^{-\pi/2\zeta}\right)}\right) \sim \frac{\pi}{48}\zeta^{-1}\,, \tag{56}$$

reproducing Equation (47) for both boundary conditions.

Note that in the anisotropic case (as with the calculations of [19]) the predictions of conformal field theory are reproduced when the aspect ratio is rescaled by a factor $\Xi$, corresponding to the observation that a conformally invariant scaling limit can only be obtained only after a corresponding anisotropic rescaling.

## Acknowledgments

This paper arose out of discussions with Alessandro Giuliani.

**Funding information**  The work was supported by the MIUR Excellence Department Project MatMod@TOV awarded to the Department of Mathematics, University of Rome Tor Vergata, CUP E83C23000330006.

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
