# Peer review of "The boundary disorder correlation for the Ising model on a cylinder"

_SciPost Physics Core, doi:SciPost Phys. Core 8, 017 (2025)_

## Round 2 · Referee Report · Anonymous (Referee 1) · 2024-8-25

Strengths

1- The results are interesting and, to the best of my knowledge, new. 2- The results provide a nice addition to the existing literature on finite size scaling for the Ising model.

Weaknesses

1- The presentation of the results can be improved.

Report

I believe the paper can be accepted after a suitable revision (see the attached file).

Requested changes

See the attached file.

Attachment

Recommendation

Ask for minor revision

---

## Round 2 · Referee Report · Anonymous (Referee 2) · 2024-11-22

Strengths

1) New results for the expression on the finite scaling of the Ising model in a two dimensional cylindrical.

Weaknesses

1) Style of writing 2) No proper introduction 3) Inconsistent notations 4) The results are new but minor extensions of the theory

Report

The paper presents new results about the correlation function of disorder insertions for a cylindrical Ising model at criticality with open and periodic (anti-periodic) bound- ary conditions. Following the theory of Kadanoff-Ceva for disorder operators, the corre- lation can be written indeed in terms of the ratio of the partition function with defects and without. In particular, the ratio considered is between the partition function with open/periodic boundary condition and the partition function with open/antiperiodic boundary conditions. Finite size corrections to the pressure are provided and shown to be consistent with CFT’s predictions.

Despite the results are new, I found the article difficult to read and in several places imprecise. Starting from the introduction, it seems confused and without a solid structure. The paper starts with ”I study the disorder correlation” without even explaining the notations of formulas (1) and (2). The author should put this results in a more general context. For instance, there should be a clearer link to Kadanoff- Ceva theory and the fermionic interpretation. Moreover,in the introduction formulas are given without explaining the variables. All this makes the article very hard to read.

Furthermore, I have the perplexities about the novelty of the paper. The core of the article is the asymptotics of the partition function using the McCoy-Wu solution for the Ising model in cylindrical boundary conditions. The author already reviewed this method (in a simplified form) in the (clearer) paper The Ising model on a cylinder: universal finite size corrections and diagonalized action, where the exposition is more formal and less sketchy. The results of Sections 2,3 are straightforwards generalizations to the anisotropic case and to the different boundary conditions of Sections 3 and 4 of the mentioned paper.

Hence, I would not recommend the publication of the submitted article. However, since the manuscript I have mentioned (i.e., The Ising model on a cylinder: universal finite size corrections and diagonalized action) seems to appear only in the arxiv, a well written review of the finite scaling of the Ising model on a cylinder with general boundary conditions (including these new results) might deserve to be published.

Requested changes

  1. Explanations about the quantities in formulas (1)-(6) are due.
  2. Explain the different notations for the θ functions: θk(·,·) and θk(·|·) in formula (6).
  3. In formula (7) I would emphasize that p, s are constants that do not depend on the boundary conditions.
  4. There is a different notation used in formula (2) and in formulas (10)-(15): M, N instead of M, N .
  5. I think formula (10) is wrong if the partition function is as in (2): I expect a 4MN in the exponent instead of MN and 2N(M − 1 instead of N(M − 1).
  6. The quantity t in the matrices (12) and (14) it is not introduced: is it the critical temperature of the system or zi?
  7. In the text after formula (15) there should be product instead of sum.
  8. The description for the matrices B(θ) is too vague. If they are really the same as of the ones in equations (3.6a) and (3.6b) of the reference [23], why the notation has an explicit dependence on the boundary conditions?
  9. The sentence after equation (44) is also too vague: which other expressions? Please mention them. Moreover, please show how you get equation (6) in more details.
  10. I would not call Section 4 Conclusions but rather something like Comparisons with CFT’s predictions.
  11. I would also recall results for the predictions for general cylindrical symmetry, e.g., π/24 for open boundary conditions.

Attachment

Recommendation

Reject

---

## Round 3 · Referee Report · Anonymous (Referee 2) · 2025-1-25

Report

The revised version of the manuscript has tackled most of my perplexities, so that I can propose the acceptance of the paper.

Recommendation

Publish (meets expectations and criteria for this Journal)

---

## Round 3 · Author Response

In response to the referee reports I have made a number of changes throughout the manuscript. I recognize that there were an unusual quantity of errors and problems of presentation, which I sincerely regret. I hope that these are corrected in the new version.

That said, one of the referees' recommendations was mainly based on an assessment of the relevance of the main result. I am afraid that in this respect I am only able to restate the arguments in the paper (that this disorder correlation plays an interesting role in the exact solution for boundary spin correlations on the cylinder), which I hope I have made clearer in the present version. I am aware that the significance of the new result is modest, and I do not wish to exaggerate it, and I do not feel competent to expand it into a full review of similar asymptotic expansions. The only consideration I would add regarding the originality of the work is that it incorporates (with some corrections) the most original part of Reference [18], an unpublished preprint which I posted in 2014 (most of the rest of that preprint involves an alternative to part of the McCoy-Wu solution which is unnecessary for the calculation of the partition function but turned out to be helpful for the study of the fermionic correlation functions in [4]).

Nonetheless, I think that the revisions provide a substantial improvement over the original version, and I hope that they meet with your approval. If not, I still wish to thank the referees for taking time and effort to review the manuscript and for helping me to improve it (whether or not it is destined for formal publication).

---

## Round 3 · List of Changes

I have made a number of changes to the introduction to clarify the notation (including adding Figure 1 and changing the symbol for the rescaling factor to Ξ rather than ξ to make it easier to distinguish from ζ), and attempted to clarify the discussion regarding the disorder correlation and its relationship to other correlation functions (including adding references [1-2,6] and updating [5]).
- I have also added a further explanation of the notation for the theta functions after Equation (6), - made a grammatical correction in the citation of [18] on page 3, - and added to the text after Equation (7) to state more clearly that p,s do not depend on the boundary conditions (among the two cases under consideration).

In Section 2, I have changed the discussion of the McCoy-Wu solution to eliminate a number of details which were not relevant (as well as being poorly explained) and instead indicate directly the point of departure for the analysis. I have also added a passage around Equation (10) to explain that the critical values of z₁ and z₂ depend only on the ratio E₁/E₂ of the coupling parameters (explaining the seeming paradox that the rescaling of the aspect ratio is expressed in terms of only z₁ in Equation (14)).
- I have also eliminated some unexplained and confusing notation (M=2𝓜, N=2𝓝, and t in place of either z₁ or z₂) that I had carelessly used here and - made a minor correction in the text between Equations (11) and (12).

I have made a number of corrections and clarifications in Section 3. In particular: - made it clearer in the text before equation (18) which limit is being considered - explained in more detail the origin of the coefficients expressed in (30) - clarified that (33) is the contribution of non-small values of θ, so that only small values remain to be controlled - added more explicit cross references and Equation (45) to explain the passage from Equation (45) to Equation (6)

In Section 4, I have changed the title as suggested by the second referee, and attempted to improve the discussion at the beginning of the section by more clearly stating the predictions of [11] which are to be compared to my results.
- I also recall the restriction 𝓜/𝓝 -> 0 (between Equations (47) and (48)), and - correct my erroneous use of x₊, x₋ in place of z₊, z₋.

---

## Editorial Decision

published